# Spatial Heterogeneity of Root Water Conduction Strategies of Zygophyllaceae Plants in Arid Regions of China

**DOI:** 10.3390/biology11101502

**Published:** 2022-10-13

**Authors:** Ying Chen, Yanjun Dong, Jie Liu, Zongshan Li, Xiaochun Wang, Maierdang Keyimu, Cong Wang, Guangyao Gao, Xiaoming Feng

**Affiliations:** 1Center for Ecological Research, Northeast Forestry University, Harbin 150040, China; 2State Key Laboratory of Urban and Regional Ecology, Research Center for Eco-Environmental Science, Chinese Academy of Sciences, Beijing 100085, China; 3National Observation and Research Station of Earth Critical Zone on the Loess Plateau in Shaanxi, Xi’an 710061, China; 4State Key Laboratory of Desert and Oasis Ecology, Xinjiang Institute of Ecology and Geography, Chinese Academy of Sciences, Urumqi 830011, China; 5Xinjiang Key Laboratory of Desert Plant Roots Ecology and Vegetation Restoration, Xinjiang Institute of Ecology and Geography, Chinese Academy of Sciences, Urumqi 830011, China

**Keywords:** arid and semi-arid areas, Zygophyllaceae plants, root anatomy, life history strategy

## Abstract

**Simple Summary:**

Desert plants are the main body of species diversity in desert ecosystems and often employ different strategies to adapt to their surrounding environment. In this study, the water conduction strategies of Zygophyllaceae plants along the drought gradient were discussed by dissecting the roots of Zygophyllaceae plants based on their vessel characteristic parameters. The results show that: (1) With the aggravation of drought stress, the water use strategy of Zygophyllaceae plants’ roots changed from safety priority to efficiency priority. (2) The root age of Zygophyllaceae plants increased with temperature, and the growth rate showed a downward trend. (3) Altitude mainly influenced plant growth by affecting the temperature and precipitation of arid habitats. The root xylem anatomical characteristics and life history strategies of Zygophyllaceae plants elaborated in the present study may provide a scientific basis for the ecological restoration of vegetation in arid and semi-arid areas of China.

**Abstract:**

Desert plants are the main component of species diversity in desert ecosystems, and studying the anatomy and function of desert plant xylem is of great significance for understanding climate sensitivity and adaptation mechanisms to arid ecosystems. In this study, 11 sampling points were selected in the region starting from the Loess Plateau to the Jungar Basin, the taproot anatomy materials of 9 samples of Zygophyllaceae plants were collected, and the water conduction strategies and spatial distribution characteristics of these species were analyzed. The age, growth rate, vessel number, vessel fraction, vessel area within a fixed measurement range (TVA), MVA, water conductivity (TKp, MKp) and vessel diameter ranged between 1 and 27 years, 43.67 and 678.10 μm/year, 20 and 1952, 4.43 and 26.58%, 8009.62 and 192069.12 μm², 27.63 and 2164.33 μm², 0.417 and 364.97 kg m^−1^ MPa^−1^ s^−1^, 0.000624 and 7.60 kg m^−1^ Mpa^−1^ s^−1^, and 5.57 and 73.87 μm, respectively. The number of root vessels (*R* = 0.27, *p* > 0.05) of Zygophyllaceae plants decreased with the decrease in precipitation, and the average vessel area (*R* = −0.28, *p* > 0.05) and hydraulic diameter (*R* = −0.29, *p* > 0.05) showed an upward trend. This shows that the water hydraulic efficiency priority strategy may be adopted in the root system of Zygophyllaceae plants in severe drought stress condition, and the water hydraulic safety priority strategy may be used in mild drought stress conditions. With the increase in temperature, the root age of Zygophyllaceae plants showed an increasing trend, and the growth rate showed a downward trend, indicating that the radial growth of the roots of Zygophyllaceae plants is mainly affected by temperature. Altitude influences plant growth by affecting temperature and precipitation in arid habitats. The findings of the present study on root xylem anatomical characteristics and life history strategies provides a scientific basis for the ecological restoration of vegetation in arid and semi-arid areas of China.

## 1. Introduction

Drought stress affects plant stability, structure, function, and carbon sinks [1]. Climate change and its effect on plant phenomena has become a major concern among researchers. Plants mainly rely on roots to absorb water from the soil, providing physiological activities and transpiration such as plant growth and development, metabolism, and so on. The water absorption of plants depends on the intrinsic water transport capacity of the root system, namely, the root hydraulic power. In general, water is rapidly transported from the soil to the root column cells and then through the xylem vessels to the aboveground part of the plants [2]. However, vessels are water guide structures in angiosperms, and vessel size and density change under different drought stress conditions [3]. Serious xylem caved embolization (xylem embolism) can affect the effectiveness and safety of plant water transport, thus restricting the normal growth and development of plants. Studies have shown that different plants respond differently to drought stress, with *Nitraria tangutorum* adapting to drought stress habitats through a relatively stable spatial pattern [4,5]. In contrast, *Tetraena mongolica* is characterized by low photosynthesis, low transpiration and high water use efficiency under drought stress [6], and *Sarcozygium xanthoxylon* enhances plant adversity adaptability by lifting water to alleviate plant water deficits [7], which indicates that plants are increasingly capable of adapting to drought-stressed habitats in the context of a warming climate. The anatomical characteristics of water conductivity in the xylems of plant roots (e.g., vessel size, the vessel number and density) reflect the effectiveness and safety of plant water transport and reveal both the internal mechanism and the adaptation mechanism of plants to the external environment [8]. These values serve as a comprehensive indicator to evaluate the adaptability of plants to drought stress [9].

In recent years, many studies have explored the response of plants to drought-stressed habitats from different perspectives. However, phenological changes in vegetation and their response to climate are mainly discussed through remote sensing and positioning experimental methods, such as the following: (1) studies based on remote sensing vegetation time series data [10]; (2) vegetation cover change on the Loess Plateau and its response to climate change based on the Improved CEEMDAN method [11]; (3) GIMMS NDVI3 g.v1 data from 1982 to 2015, combined with site meteorological data, spatial and temporal characteristics, climatic response and future trends of vegetation cover in the Horqin Sandy Land, discussed by trend analysis, coefficient of variation, Hurst index and partial correlation analysis. The results showed that precipitation was the main factor affecting the vegetation changes observed in the study area [12]. Using a simulated precipitation control experiment, the response of the hydraulic characteristics of *Salix psammophila* and *Caragana korshinskii* stems of typical shrubs to the simulated precipitation change in the water erosion and wind erosion staggered zone of the Loess Plateau was studied [13]. In the context of drought stress caused by global warming, the existing information on the anatomical characteristics of plant xylems is not sufficient. Previous research focused mainly on the characteristics of the radial growth of trees in response to the climate, and knowledge on the anatomy of plant root water conduction is still relatively limited; thus, it is necessary to study the characteristics of the climate response of the root water conduction structure of plants of the same family in different regions, as these results can help us understand the life history strategy of plants under stressed habitat conditions and more accurately predict the dynamic trends of plants under future drought stress scenarios.

Frequent drought, scanty precipitation, and high evapotranspiration are typical climatic characteristics of arid and semi-arid regions of China. This resource-scarce situation makes these regions more ecologically fragile in light of present global changes [14]. The study of vegetation response characteristics to climate has become the basic guiding work in the reconstruction of the ecological environment [15]. In this study, a large number of root samples of Zygophyllaceae plants were obtained along the precipitation gradient of the arid and semi-arid regions of China (Hexi Corridor–Jungar Basin).

In this study, the parameters of root anatomical characteristics (including the number of vessels, vessel size, vessel area, and water conduction efficiency) were extracted. The objectives are: (1) to examine the anatomical characteristics of the root vessel of Zygophyllaceae plants growing in the arid and semi-arid regions of northwest China; (2) to explore the variation trend of the water conduction efficiency of Zygophyllaceae plants with the changes in precipitation, temperature and altitude; and (3) to identify the hydraulic strategies of Zygophyllaceae plants studied in a drought stress environment. The characteristics of the water conductivity of Zygophyllaceae plants in the region were elucidated along the gradient of drought stress, which provides basic data for the further study of the response of ecosystems to climate change and the restoration of vegetation in arid and semi-arid areas.

## 2. Materials and Methods

### 2.1. Overview of the Study Area and Sample Collection

The study area is located in the arid and semi-arid region of northern China (Figure 1). It mainly includes the Hexi Corridor (located at the confluence of the three major plateaus of Loess, Qinghai–Tibet and Mongolia, with complex terrain, crisscrossed by mountains and vast differences in altitude), and the Jungar Basin (the northern part of Xinjiang, China, is the second largest inland basin in China, located between the Altai Mountains and the Tianshan Mountains, on the west side of the Jungar–Altai Mountains, and east of the foothills of the Beita Mountains). Its geographical location is 37°22′–45°08′ N, 88°34′–107°37′ E and belongs to a temperate continental monsoon climate. The sampling points range from southeast to northwest, and the climate is semi-humid, semi-arid, and arid. The annual precipitation has gradually decreased, the dryness level has gradually increased, and the annual sunshine hours have increased from east to west [16]. The altitude gradient ranges from 800 to 1793 m. The altitudes of different sampling points are quite different, the annual average temperature is 3.65–10.44 ℃; the winter is severe, the summer is hot, and the annual temperature difference is large. The annual precipitation ranges from 45.27 to 278.01 mm, and the annual precipitation is mainly concentrated from June to September, so the growth and reproduction of most plant species mainly depend on the atmospheric precipitation in this period. Typical vegetation species mainly include *Peganum harmala*, *Nitraria tangutorum*, *Tribulus terrestris*, *Sarcozygium xanthoxylon*, and *Calligonum mongolicum* (Figure 1, Table 1).

According to the specific conditions of the test plot and the needs of the test, topography and species distribution were considered under the conditions of flat terrain, and there were obvious characteristics of species community structure and no strong interference. We collected a total of nine species of Zygophyllaceae plants from Yanchi (44°22′ N, 107°24′ E) to Dabancheng (43°24′ N, 88°34′ E) in July and August from 2020 to 2021 (Figure 1, Table 1). Each species selected three to five healthy and mature plants, and their taproots were dug and cleaned (Figure 2). Root samples 2–3 cm (Figure 2) beneath the ground surface were cut with blades and fixed in FAA fix fluid (70% ethanol:formaldehyde:acetic acid = 9:0.5:0.5) more than 50 times.

### 2.2. Sample Processing

The collected plant samples were pre-treated according to the experiment protocol, i.e., dehydration, transparency, wax immersion, embedding, sectioning, dyeing and observation. Dehydration involves removing the plant samples from the fixative solution, rinsing them with distilled water, and successively placing them in different concentrations (80%, 85%, 90%, 95%, 100% I. and 100% II.) of the ethanol solution.

Transparency is the use of xylene as a medium to replace the ethanol solution in the dehydrated tissue, which is conducive to late wax immersion. Dip wax involves the use of paraffin instead of a clear agent to immerse paraffin into plant tissue, which plays a supporting role when sectioning. Embedding is the process of wrapping a wax-soaked sample in a paraffin solution and solidifying the sample at room temperature. The slice is used to cut out the solidified wax block with a rotary microtome of approximately 8–12 µm thickness of the cross-sectional wax strip; the wax strip is placed on the slide and flattened completely with distilled water, and then the slide is placed in the 45 ℃ dryer for more than 24 h. The dyeing is carried out by the crocus solid green dyeing method, and in terms of vascular bundles, the xylem is dyed red by safranine (not easy to color), and the phloem is dyed green by fast green (easy to color). After dyeing, the tablets were sealed with neutral gum, placed in a fume hood to dry, and were used to prepare the paraffin sections [17,18,19]. Finally, an Olympus DP73 (Olympus, Tokyo, Japan) was used to take images at the same magnification (×40).

### 2.3. Data Processing and Statistical Analysis

Using ImageJ’s software, a fixed area (425 µm × 425 µm) on the slice was selected as the area used to measure the xylem vessel (Figure 3), the anatomical characteristics of the water conduction of the measured section were automatically read, the parameter values were obtained (Figure 1), and the following vessel parameters were obtained. (1) The number of vessels (NV): the number of vessels within the measurement range. (2) Vessel fraction (VF) (%): the proportion of the total conduit area in the measurement range to the measured area. (3) Total vessel area (TVA) (μm²): the total area of the vessel in the measurement area. (4) Mean vessel area (MVA) (μm²): the average of the total area of all vessels in the measurement range. (5) Total hydraulic conductivity (TKp) (kg m^−1^ MPa^−1^ s^−1^): the sum of the hydraulic conductivity efficiencies in the measurement area. 6) Mean hydraulic conductivity (MKp) (kg m^−1^ MPa^−1^ s^−1^): the average water conduction efficiency of the vessels in the measurement area.

Since the vessels are not perfectly round but are approximately elliptical, each vessel (equivalent circle) diameter is calculated using the formula:$$D\;={\left[3{\left(\mathrm{ab}\right)}^{3}/\left(2a^{2}+2b^{2}\right)\right]}^{1/4}$$
where a and b are the values of the major and minor axes of the vessel, respectively [20]. According to the Hagen–Poiseuille equation, the theoretical hydraulic conductivity (Kp)
$$\mathrm{Kh}\;=\left(\pi \rho /128\eta \right)\overset{n}{\underset{𝒾=1}{\sum }}{\left(D_{𝒾}\right)}^{4}$$

Can be calculated, where *p* is the density of water at 20 ℃ (998.2 kg/m^3^), and the viscosity of water at 20 ℃ (1.002 × 10–9 MPas) and the diameter of the first vessel (equivalent circle) measured in the first year are used. According to the formula, the hydraulic diameter (DH)
$$\mathrm{DH}=\overset{n}{\underset{𝒾=1}{\sum }}{\left(D_{𝒾}\right)}^{5}/\overset{n}{\underset{𝒾=1}{\sum }}{\left(D_{𝒾}\right)}^{4}$$
is calculated using the diameter of each equivalent circle, where the diameter of the first vessel (equivalent circle) is measured in the first year [21,22].

The relevant data in this study were recorded and organized using WPS Office. The data were mainly managed with SPSS 23.0 (IMB SPSS Statistics 23.lnk, https://www.ibm.com/cn-zh/spss) software and the R language program for Pearson correlation analysis and probability analysis; graphics were drawn using the R language program, SPSS 23.0 (IMB SPSS Statistics 23.lnk), and Sigmaplot14.0 and ArcGIS 10.2 software were used for completion.

## 3. Results and Analysis

### 3.1. Change in the Vessel Characteristic Parameter Value of Zygophyllaceae Plants

In this study, a total of 9 species of Zygophyllaceae plants were collected from 11 sampling sites in the arid and semi-arid regions of northern China, belonging to 6 genera (Table 2). Of these, the largest number of species was from the camel petal genus (up to 4 species), namely, *Zygophyllum fabago*, *Zygophyllum mucronatum*, *Zygophyllum gobicum*, and *Zygophyllum potaninii*. All other genera had only one species, namely, *Nitraria tangutorum*, *Sarcozygium xanthoxylon*, *Artemisia brachyloba*, *Peganum harmala*, and *Tribulus terrestris*. Among them, *Sarcozygium xanthoxylon* and *Nitraria tangutorum* are shrubs, *Artemisia brachyloba* is a semi-shrub-like herb, *Tribulus terrestris* is an annual herb, and other species are perennial herbs.

From the perspective of the parameter frequency characteristics of the root vessel characteristics of Zygophyllaceae plants (Figure 4), the age (age), vessel fraction (VF), average vessel area (MVA), total water conduction efficiency (TKp), average water conduction efficiency (MKp), and hydraulic diameter (DH) all showed normal distribution characteristics, and the number of vessels (NV), growth rate (Growth.rate) and total vessel area (TVA) showed normal distribution characteristics. The age, the vessel number and vessel fraction distribution were more concentrated. The medium age (5–8 years) was dominant, of which the age of white thorn was the oldest at 27 years. The vessel number was mainly concentrated between 110 and 509, and the smallest and largest vessel numbers were found for *Peganum harmala* and *Zygophyllum fabago*, with values of 20 and 1952. The vessel score was concentrated between 8% and 18%, and the smallest and largest vessel score species values were found for white thorns, with values of 4.40% and 26.60%. The growth rate was mainly low (43.60–165.80 µm/year), and the species with the largest growth rates were the large-flowered camel hoof and *Tribulus terrestris*, with values of 658.10 µm/year and 678.10 µm/year. The species with the smallest and largest total vessel areas were camel pony and white thorn, with values of 8009 μm² and 192,069 μm². The distributions of average duct area, average hydraulic conduction efficiency and hydraulic diameter were more dispersed, but they were clearly concentrated on the peak, with peak range values of 155–300 μm², 0.03–0.08 kg m^−1^ MPa^−1^ s^−1^, and 18–23 µm, respectively, and the average vessel area, average hydraulic conduction efficiency and hydraulic diameter of the smallest and largest species were found for camel pony. The values were 27.60 μm², 0.0006 kg m^−1^ MPa^−1^ s^−1^, 5.60 µm, 2164.30 μm², 7.60 kg m^−1^ MPa^−1^ s^−1^, and 73.90 µm, respectively. The total water conductivity was dominated by medium-to-low values (4–24 kg m^−1^ MPa^−1^ s^−1^ species), while there were fewer species which had low and high total hydraulic conductivity efficiency, of which the lowest and highest total water conductivity species was camel pony, with values of 0.42 kg m^−1^ MPa^−1^ s^−1^ and 365.00 kg m^−1^ MPa^−1^ s^−1^.

From the correlation between the parameters of root vessel characteristics of Zygophyllaceae plants (Figure 5), the vessel number was significantly negatively correlated with the mean duct area, hydraulic diameter and average hydraulic conduction efficiency (*R* = −0.74–−0.83, *p* < 0.01), which was significantly negatively correlated with the total hydraulic conductivity (*R* = −0.52, *p* < 0.05), but there was no obvious correlation with height, age, total vessel area, or growth rate. The growth rate was significantly negatively correlated with age (*R* = −0.52, *p* < 0.05). The average hydraulic conduction efficiency was significantly positively correlated with the total hydraulic conduction efficiency, average duct area, and hydraulic diameter (*R* = 0.91–0.97, *p* < 0.01). The hydraulic diameter was significantly positively correlated with the total hydraulic conduction efficiency and average duct area (*R* = 0.92–0.98, *p* < 0.01). The mean duct area was significantly positively correlated with the total hydraulic conduction efficiency (*R* = 0.89, *p* < 0.01), and the total hydraulic conduction efficiency was significantly positively correlated with the total duct area and vessel fraction.

### 3.2. Change Law of the Vessel Structure Parameter on the Precipitation Gradient of Zygophyllaceae Plants

From the perspective of the change trend of the root vessel characteristic parameters with the precipitation gradient of Zygophyllaceae plants (Figure 6), the root age, average vessel area, hydraulic conduction efficiency (TKp, MKp) and hydraulic diameter all showed a continuous downward trend with the increase in precipitation, of which the most obvious downward trend was detected for age and hydraulic diameter. On the precipitation gradient of 45.27–278.01 mm, the age and hydraulic diameter decreased from 27 years and 73.90 µm to 1 year and 5.60 µm, respectively. The growth rate, vessel fraction and total vessel area fluctuated steadily with the change in precipitation, and there was no obvious change trend. The growth rate, vessel fraction and total vessel area were basically maintained at 200 µm/year, 13%, and 78,500 µm^2^, respectively. The vessel number and the increase in precipitation showed a continuous upward trend, and the vessel number increased from 20 to 1952 along the precipitation gradient of 45.27–278.01 mm.

### 3.3. Variation in Vessel Structure Parameters in Temperature Gradients of Zygophyllaceae Plants

From the perspective of the trend of root vessel characteristic parameters with the temperature gradient of Zygophyllaceae plants (Figure 7), the root age, vessel area (TVA, MVA), and hydraulic diameter all showed a continuous upward trend with temperature, of which the most obvious upward trend was found for the total vessel area (*R* = 0.63, *p* < 0.01), which spanned from 70 at 3.60 °C to 260 at 10.40 °C. The growth rate and vessel fraction decreased with increasing temperature, from 270.40 µm/year and 13.30% at 3.60 °C to 68.30 µm/year and 10.50% at 10.40 °C, respectively, and the vessel number and hydraulic conduction efficiency (TKP, MKp) did not change significantly with temperature, and basically remained stable at 430,47 kg m^−1^ MPa^−1^ s^−1^, and 0.62 kg m^−1^ MPa^−1^ s^−1^.

### 3.4. Variation in Vessel Structure Parameters along the Elevation Gradient of Zygophyllaceae Plants

From the perspective of the change trend of root vessel characteristic parameters with the altitude gradient of Zygophyllaceae plants (Figure 8), the root age, growth rate, vessel fraction, average vessel area, water conduction efficiency (TKp, MKp), and hydraulic diameter all showed a continuous downward trend with the increase in altitude, of which the most obvious downward trend was found for age and average water conduction efficiency, with values that decreased from 10 years at 800 m and 0.79 kg m^−1^ MPa^−1^ s^−1^ to 5 years at 1793 m and 0.01 kg m^−1^ MPa^−1^ s^−1^, respectively. The vessel number and the total area of vessel increased with increasing altitude, and the vessel number (*R* = 0.48, *p* < 0.05) and the total vessel area increased from 20 and 8009 μm² at 800 m to 1952 and 128,091 μm² at 1793 m, respectively.

## 4. Discussion

### 4.1. Basic Performance Characteristics of Root Vessels of Zygophyllaceae Plants in Arid and Semi-Arid Areas in China

Due to the wide area of the sampling region from the Loess Plateau along the Hexi Corridor to the Jungar Basin (37°22′−45°08′ N, 88°34′−107°37′ E), the climatic conditions were diverse, and the values of the root vessel parameters of Zygophyllaceae plants in the region also had a large variation range. The vessel number (NV), the fraction of the vessels (VF) and the average vessel area (MVA) in the fixed measurement area were 20–1952, 4.40–26.60%, 27.60–2164.30 μm², respectively. The diameter of the root vessel was 25.23 ± 3.81 µm, and the diameter of the root vessel was lower than that of the drought-tolerant tree species of *Robinia pseudoacacia* (56.96 ± 1.92 µm), *Ulmus pumila* (49.48 ± 1.21 µm), *Salix matsudana* (32.11 ± 0.42 µm), *Acer truncatum* (31.26 ± 0.41 µm), *Corylus heterophylla* (26.81 ± 0.32 µm) [23], and *Tetraena mongolica* (46.91 ± 6.71 µm) [24]. The Zygophyllaceae plants studied had smaller duct diameters than those of *Robinia pseudoacacia, Acer truncatum, and Tetraena mongolica*, which was mainly due to the more stressed growth conditions faced by Zygophyllaceae plants. Narrow and thick-walled vessel have low hydraulic conductivity but are not easy to lodge, which can enhance the strength of the vessel and resist the high negative pressure generated by the arid environment. For Zygophyllaceae plants growing in areas with low annual precipitation, the vessel diameter is small, which is conducive to enhance the ability to resist negative pressure in the body [25]. The second possible reason is that most of the species of *Zygophyllaceae plants* are herbaceous plants, while the drought-tolerant tree species studied by Li Rong and Jiang Sha were mainly shrubs.

### 4.2. Characteristics of the Root Ducts of Zygophyllaceae Plants in Arid and Semi-Arid Areas of China Change along the Precipitation Gradient

The average root area, water conductivity (TKp, MKp), and hydraulic diameter of Zygophyllaceae plants in arid and semi-arid regions of China showed a continuous downward trend with increasing precipitation (Figure 5), and the average root duct area, hydraulic conduction efficiency (TKp, MKp), and hydraulic diameter ranged from 292.44 μm^2^, 13.00 kg m^−1^ MPa^−1^ s^−1^, 0.05 kg m^−1^ MPa^−1^ s^−1^, 0.05 kg m^−1^ MPa^−1^ s^−1^, and 19.87µm to 248.36 μm^2^, 9.47 kg m^−1^ MPa^−1^ s^−1^, 0.03 kg m^−1^ MPa^−1^ s^−1^, and 18.39 µm. The vessel number showed an upward trend with increasing precipitation, and the vessel number increased from 20 to 1952 on the precipitation gradient ranging from 45.27 mm to 278.01 mm. This result shows that the hydraulic conduction strategy of Zygophyllaceae plants in arid and semi-arid areas of China has undergone significant changes along the precipitation gradient. In relatively arid areas, the root duct area of Zygophyllaceae plants is large, the vessel number is small, and the hydraulic conductivity efficiency is high. In relatively humid areas, the rhizome and leaf vessel area of Zygophyllaceae plants is small, the vessel number is large, and the hydraulic conduction efficiency is low. This result is consistent with the results of Pei Tingting [26] on the sensitivity of vegetation water use efficiency to the climate and vegetation index of the Loess Plateau. Zygophyllaceae plants in relatively arid areas are characterized by drought-tolerant water use strategies [7].

In arid and water-scarce environments, water is an important factor restricting plant growth [27], and the phenotypic plasticity of vessel traits enables plants to adjust the safety and efficiency of xylem water transport to achieve efficient and safe water transport systems [28]. Plant hydraulic conduction efficiency is closely related to the diameter, density, length, and internal structure of the vessel [8]. Zimmermann believes that the effectiveness and safety of water transport is determined by the size and density of the vessel molecules [7]. When the vessel is large and the moisture conductivity is high, it is more fragile, prone to lodging, and easy to embolize [27]. When the vessel is small, the hydraulic conductivity is low, but it is resistant to negative pressure, and it is not easy to lodge and embolize. When the vessel density is large, and even if some vessels are embolized, this cannot cause the entire water transmission system to lose function, which ensures the effectiveness of water transportation. In the face of drought-stressed habitats, tree growth generally faces a trade-off between xylem hydraulic conduction efficiency and hydraulic conduction diversion safety [29]. The results of this study showed that in the arid and semi-arid areas of China, the number of vessel molecules was large (MVA, DH), the vessel number was small, the hydraulic conduction efficiency was high, and the hydraulic conduction efficiency priority strategy was adopted. In the areas with mild drought stress, the number of vessel molecules was small (MVA, DH), the vessel number was large, even if the hydraulic conduction efficiency was low, the reliability of the performance of the conduction function was guaranteed due to the high density of the vessel [30], and the hydraulic safety priority strategy was adopted.

### 4.3. Variation in Root Vessel Characteristics along the Temperature Gradient of Zygophyllaceae Plants in Arid and Semi-Arid Regions of China

The number of root vessels and the hydraulic conduction efficiency (TKP, MKp) of Zygophyllaceae plants did not change significantly with temperature (Figure 6), and values were basically maintained at 430, 47 kg m^−1^ MPa^−1^ s^−1^ and 0.62 kg m^−1^ MPa^−1^ s^−1^, respectively, indicating that temperature was not the dominant factor affecting the root hydraulic conduction efficiency of Zygophyllaceae plants. Age showed a continuous upward trend with increasing temperature, and the growth rate showed a continuous downward trend with increasing temperature. The age and growth rate ranged from 5 years and 270.40 µm/year at 3.60 °C to 68.30 µm/year and 27 years at 10.40 °C. In the past hundred years, with the continuous deepening of research, some scholars have found that with the increase in temperature, the response sensitivity of tree radial growth to climate has also changed [31], and with the increase in temperature in recent decades, Wu Xiangding [32] found that the radial growth of trees such as *Larix olgensis* and *Picea brachytyla* in the Gongga Mountains showed an upward trend. The growth rate of tree radial growth in the northeast region is faster than that in the southwest region. Han Jinsheng [33] found in Xiaoxing’anling that, after the sudden warming mutation in 1976, the radial growth of *Phellodendron amurense* increased, which was consistent with the results of this study. Warming can trigger warm drought phenomena which increase plant transpiration and water evaporation in soil or air, further exacerbate the drought stress of plants in arid habitats, inhibit plant growth, and thus show a downward trend in growth rate with increasing temperature [34].

### 4.4. Variation in the Coastal Uprooting Gradient of Root Vessel Characteristics of Zygophyllaceae Plants in Arid and Semi-Arid Areas of China

The root age and growth rate of Zygophyllaceae plants showed a downward trend with increasing altitude (Figure 8), probably because the winter temperature in high-altitude areas was low (Table 3), and the roots and branches of trees were susceptible to frost damage, affecting the photosynthesis and water absorption of trees [35], thereby restricting the growth of tree-ring width [36] and plant growth lifespan. There was a significant positive correlation between altitude and precipitation, and precipitation showed an upward trend with increasing altitude (Table 3). In this study, the vessel number showed a continuous upward trend with the increase in altitude (R = 0.48, *p* < 0.05), and the average vessel area, hydraulic conduction efficiency (TKp, MKp), and hydraulic diameter showed a continuous downward trend with the increase in altitude, of which the most obvious downward trend was found for the average water conduction efficiency, and the average water diversion efficiency decreased from 0.79 kg m^−1^ MPa^−1^ s^−1^ at 800 m to 0.01 kg m^−1^ MPa^−1^ s^−1^ at 1793 m. This result shows that in the arid and semi-arid areas of China, for Zygophyllaceae plants in the drought-stressed area, the vessel molecules are larger (MVA, DH), the vessel number is small, the hydraulic efficiency is high, and the hydraulic efficiency priority strategy is adopted. In the areas with mild drought stress, the vessel molecules are small (MVA, DH), the vessel number is large, even if the hydraulic conduction efficiency is low, and the hydraulic safety priority strategy is adopted.

## 5. Conclusions

It was concluded that the root duct characteristic parameters of Zygophyllaceae plants species in the arid and semi-arid regions of China had great variability, and the growth rate continued to decline with age, mainly due to the enhancement of the living environment stress of plants in the late growth period and the investment of more resources into reproductive growth. The average vessel area, hydraulic conduction efficiency (TKp, MKp) and hydraulic diameter of the roots of Zygophyllaceae plants showed a continuous downward trend with the increase in precipitation, and the vessel number showed a continuous upward trend with the increase in precipitation, indicating that the vessel anatomical characteristics of Zygophyllaceae plants in arid and semi-arid areas of China had strong plasticity. With the aggravation of drought stress, the hydraulic conduction strategy changed from prioritizing safety to prioritizing efficiency, and the growth rate (radial growth) showed a downward trend with temperature. The root growth rate and water conductivity of Zygophyllaceae plants showed a downward trend with increasing altitude, mainly because the altitude affected the temperature and the precipitation of the plant living environment, the decrease in temperature at high altitudes affected the radial growth of plants, and the increase in precipitation affected the hydraulic conductance capacity of plants. In this study, the vessel anatomy of the roots of Zygophyllaceae plants in arid and semi-arid areas of China and their life history strategies under arid habitat conditions were preliminarily analyzed, and the information can be used to provide a scientific basis for the reconstruction of ecological vegetation restoration in this area.

## Figures and Tables

**Figure 1 biology-11-01502-f001:**
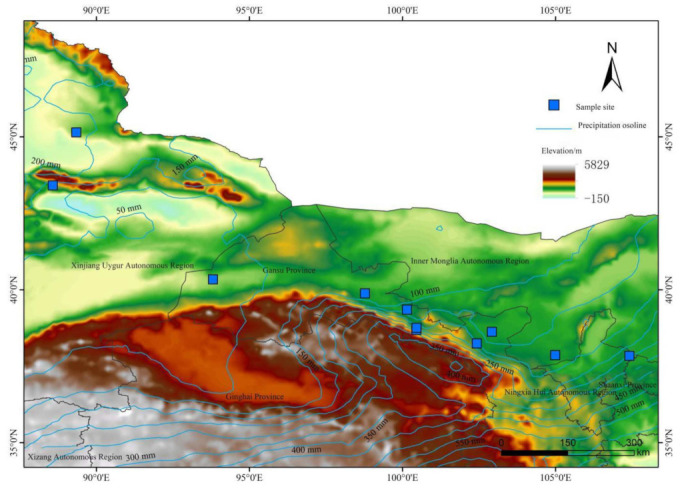
Spatial distribution of sampling points for Zygophyllaceae plants in arid and semi-arid areas of China.

**Figure 2 biology-11-01502-f002:**
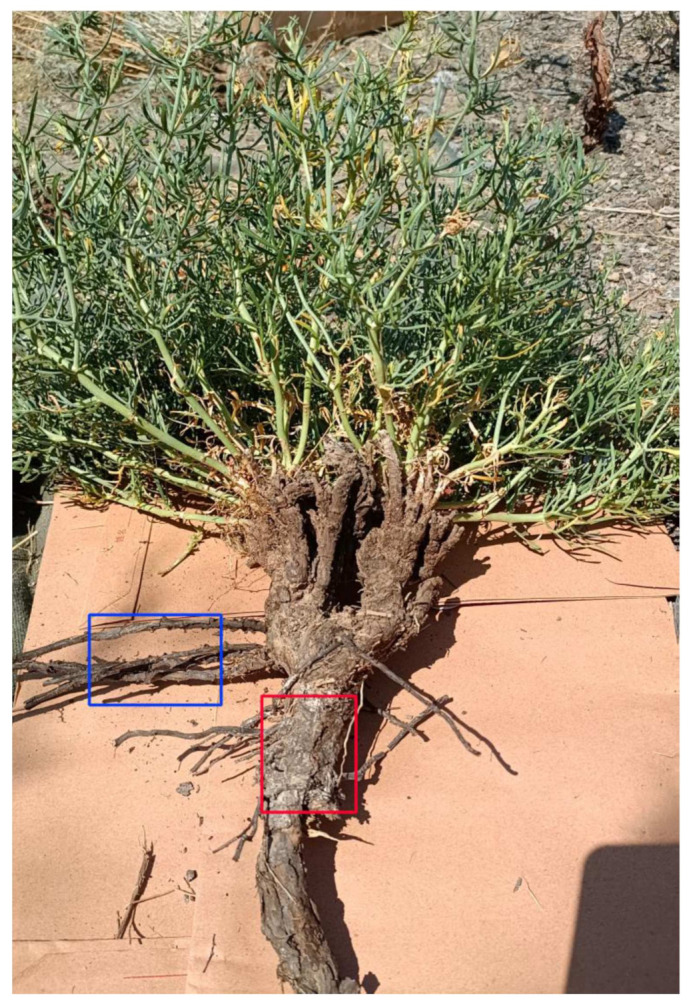
Picture of the study material (*Peganum harmala*). The red box indicates the intercepted part (taproot), and the blue box indicates the lateral roots.

**Figure 3 biology-11-01502-f003:**
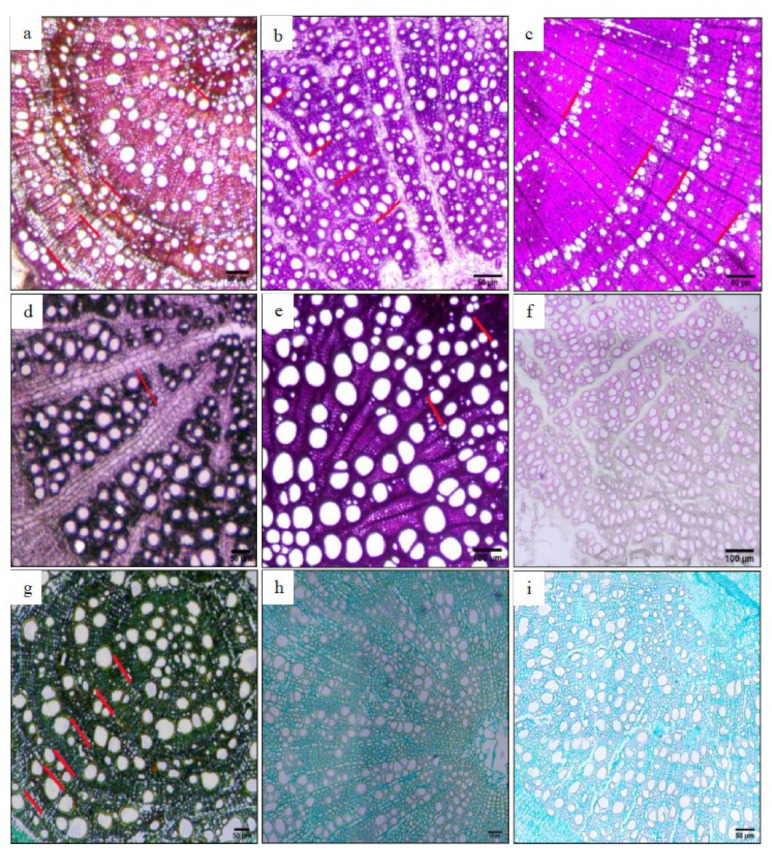
Fixed measured area and annual ring characteristics of the roots of Zygophyllaceae plants in arid and semi-arid areas of China. (**a**) *Peganum harmala*; (**b**) *Zygophyllum mucronatum*; (**c**) *Sarcozygium xanthoxylon*; (**d**) *Zygophyllum fabago*; (**e**) *Nitraria tangutorum*; (**f**) *Zygophyllum gobicum*; (**g**) *Artemisia brachyloba*; (**h**) *Tribulus terrestris*; (**i**) *Zygophyllum potaninii*. The white ellipse in the figure is the vessel.

**Figure 4 biology-11-01502-f004:**
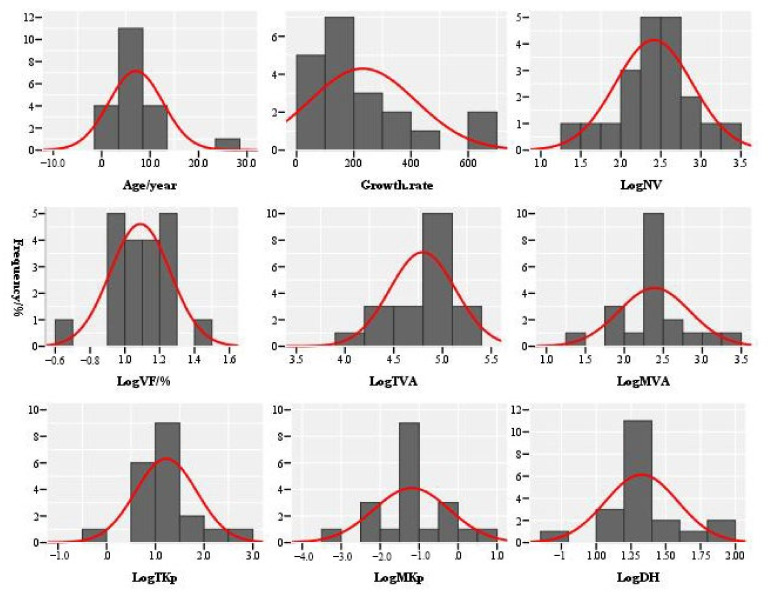
A frequency map of the root vessel characteristics of the Zygophyllaceae plants in China’s arid and semi-arid regions. Age: (year); Growth.rate: (μm/year); NV: vessel number; VF: vessel fraction (%); TAV: total vessel area (μm²); MVA: mean vessel area (μm²); TKp: total water transfer efficiency (kg m^−1^ MPa^−1^ s^−1^); MKp: mean water transfer efficiency (kg m^−1^ MPa^−1^ s^−1^); DH: hydraulic diameter (μm). Note: Before the correlation analysis, the original data were converted into log10.

**Figure 5 biology-11-01502-f005:**
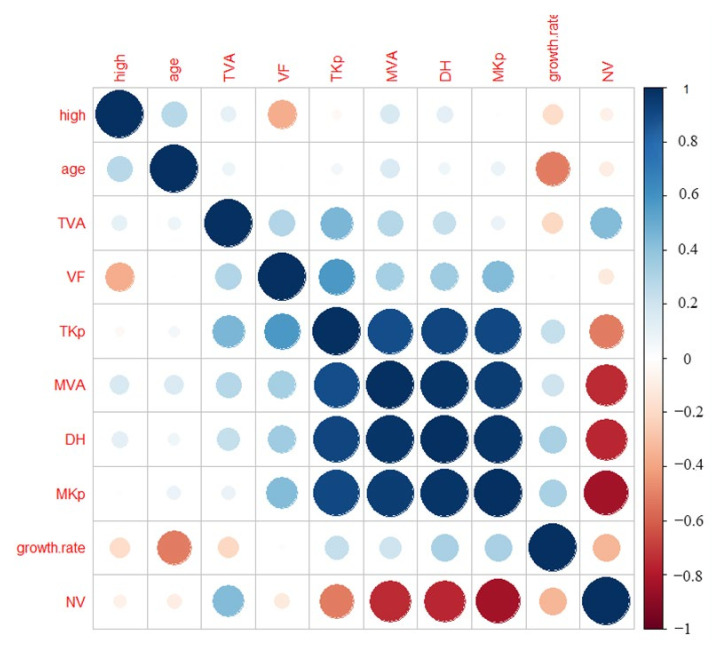
Relationship between the anatomical parameters of the roots of the Zygophyllaceae plants in arid and semi-arid areas of China. High: plant age height (m); Age: (year); Growth.rate: (μm/year); NV: vessel number; VF: vessel fraction (%); TAV: total vessel area (μm²); MVA: mean vessel area (μm²); TKp: total water transfer efficiency (kg m^−1^ MPa^−1^ s^−1^); MKp: mean water transfer efficiency (kg m^−1^ MPa^−1^ s^−1^); DH: hydraulic diameter (μm). Note: Before the correlation analysis, the original data were converted into log10.

**Figure 6 biology-11-01502-f006:**
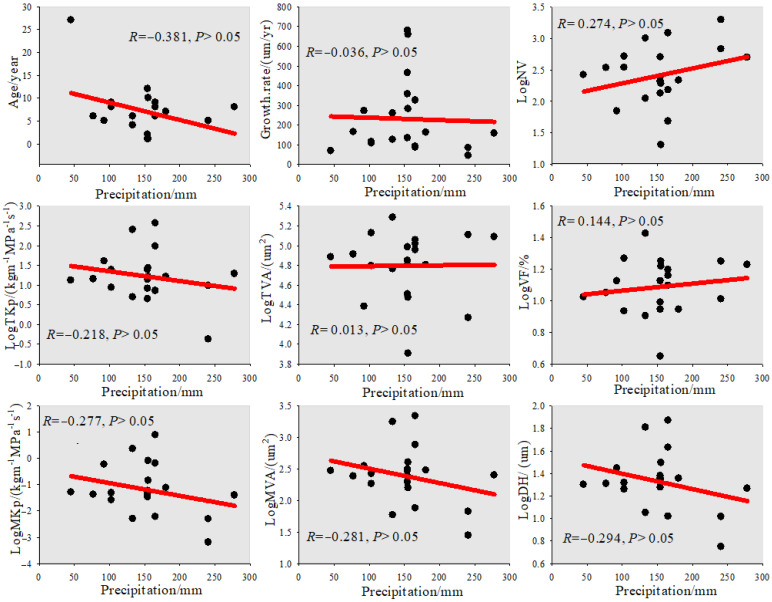
The anatomical characteristics of the root vessels of Zygophyllaceae plants in China’s arid and semi-arid regions change with precipitation gradient. Age: (year); Growth.rate: (μm/year); NV: vessel number; VF: vessel fraction (%); TAV: total vessel area (μm²); MVA: mean vessel area (μm²); TKp: total water transfer efficiency (kg m^−1^ MPa^−1^ s^−1^); MKp: mean water transfer efficiency (kg m^−1^ MPa^−1^ s^−1^); DH: hydraulic diameter (μm). Note: Before the correlation analysis, the original data were converted into log10.

**Figure 7 biology-11-01502-f007:**
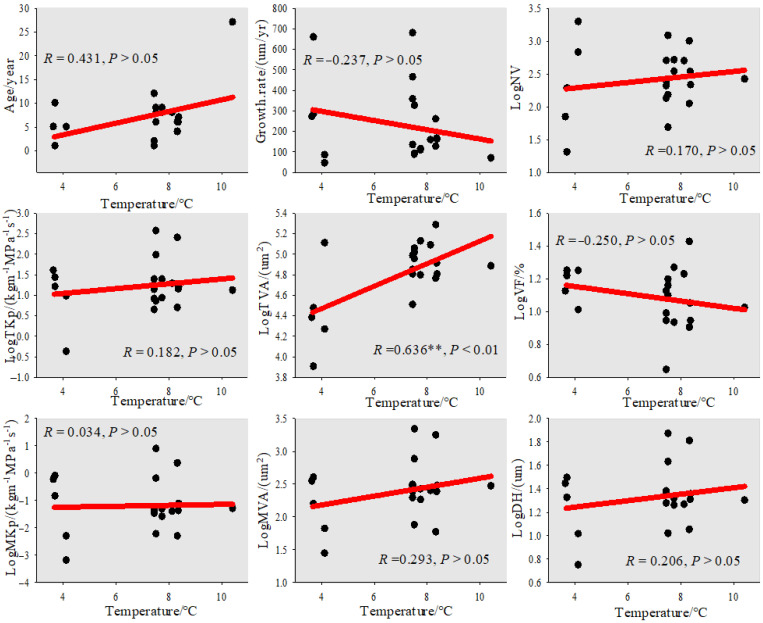
The anatomical characteristics of the root vessels of Zygophyllaceae plants in China’s arid and semi-arid regions change with temperature gradients. Age: (year); Growth.rate: (μm/year); NV: vessel number; VF: vessel fraction (%); TAV: total vessel area (μm²); MVA: mean vessel area (μm²); TKp: total water transfer efficiency (kg m^−1^ MPa^−1^ s^−1^); MKp: mean water transfer efficiency (kg m^−1^ MPa^−1^ s^−1^); DH: hydraulic diameter (μm). Note: Before the correlation analysis, the original data were converted into log10.

**Figure 8 biology-11-01502-f008:**
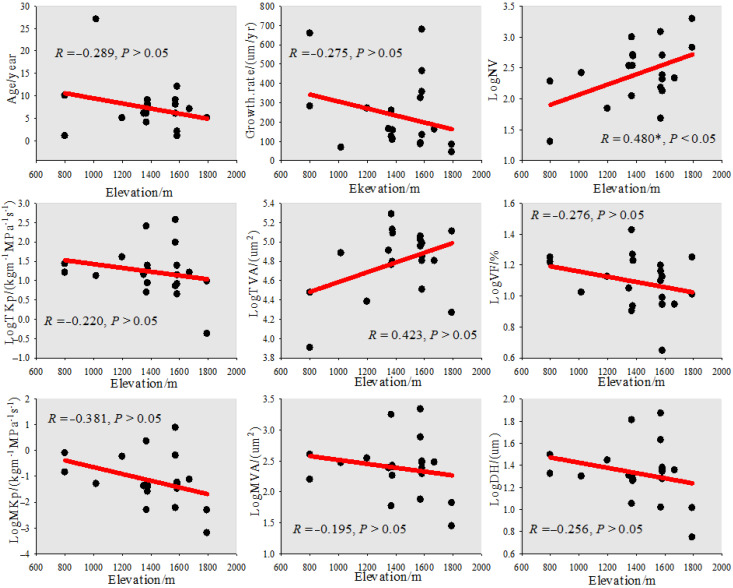
The anatomical characteristics of the root vessels of Zygophyllaceae plants in China’s arid and semi-arid regions and change in the law of altitude gradients. Age: (year); Growth.rate: (μm/year); NV: vessel number; VF: vessel fraction (%); TAV: total vessel area (μm^2^); MVA: mean vessel area (μm^2^); TKp: total water transfer efficiency (kg m^−1^ MPa^−1^ s^−1^); MKp: mean water transfer efficiency (kg m^−1^ MPa^−1^ s^−1^); DH: hydraulic diameter (μm). Note: Before the correlation analysis, the original data were converted into log10.

**Table 1 biology-11-01502-t001:** Sampling sites of Zygophyllaceae plants in arid and semi-arid regions of China.

Site ID	Site	Longitude	Latitude	Altitude (/m)	Precipitation (/mm)	Temperature (/℃)	Species	Number of Samples
S1	Yanchi xian	107°24′	37°50′	1380	278.01	8.14	Peganum harmala	5
S2	Alashan zuoqi	104°59′	37°52′	1670	180.58	8.39	Peganum harmala	5
S3	Minqin xian	102°55′	38°37′	1370	133.39	8.34	Nitraria tangutorum, Sarcozygium xanthoxylon	12
S4	Wuwei shi	102°25′	38°15′	1573	165.44	7.54	Peganum harmala, Artemisia brachyloba	12
S5	Zhangye shi	100°26′	38°41′	1793	240.88	4.14	Peganum harmala, Zygophyllum fabago	10
S6	Zhangye shi	100°08′	39°21′	1377	102.95	7.77	Nitraria tangutorum, Zygophyllum gobicum	8
S7	Zhangye shi	100°27′	38°45′	1584	154.40	7.47	Nitraria tangutorum, Zygophyllum fabago, Peganum harmala, Tribulus terrestris	16
S8	Jiuquan shi	98°46′	39°53′	1351	77.22	8.38	Nitraria tangutorum	4
S9	Dunhuang shi	93°48′	40°20′	1018	45.27	10.44	Nitraria tangutorum	3
S10	Fukang shi	89°20′	45°08′	800	155.28	3.72	Peganum harmala, Zygophyllum potaninii	7
S11	Dabancheng	88°34′	43°24′	1200	92.73	3.65	Peganum harmala	4

**Table 2 biology-11-01502-t002:** Species composition of Zygophyllaceae plants in arid and semi-arid regions of China.

Species	Genera	Life Type
*Nitraria tangutorum*	Nitraria	Shrub
*Sarcozygium xanthoxylon*	Sarcozygium	Shrub
*Artemisia brachyloba*	Artemisia	Semi-shrub-like herb
*Peganum harmala*	Peganum	Perennial herb
*Zygophyllum fabago*	Zygophyllum	Perennial herb
*Zygophyllum mucronatum*	Zygophyllum	Perennial herb
*Zygophyllum gobicum*	Zygophyllum	Perennial herb
*Zygophyllum potaninii*	Zygophyllum	Perennial herb
*Tribulus terrestris*	Tribulus	Annual herb

**Table 3 biology-11-01502-t003:** The correlation coefficients between the sampling points of Zygophyllaceae plants in arid and semi-arid areas of China.

	Elevation (m)	Precipitation (mm)	Temperature (℃)
Elevation (m)	1		
Precipitation (mm)	0.466 *	1	
Temperature (℃)	0.177	−0.346	1

Note: * At level 0.05 (two-tailed), the correlation is significant.

## Data Availability

The data presented in this study are available on request from the corresponding author. The data are not publicly available yet as the authors are writing some other papers examining these data.

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
