# Peer review of "Spatial Heterogeneity of Root Water Conduction Strategies of Zygophyllaceae Plants in Arid Regions of China"

_biology, 2022, doi:10.3390/biology11101502_

Round 1

Reviewer 1 Report

strange to note that why authors wants to incorporate too many statements in one complex big sentence and it is observed in whole document. it is strongly suggested to explain the matter in more simple small and clear sentences   

Reviewer 2 Report

Need to drastically improve the English and Grammar of the whole manuscript. 

Figures may need more elaboration. 

Figures may need to be properly labeled to highlight the cells, and paths and elaborate on the mechanism of water and root relation. 

More elaboration is needed on sampling strategies rather than just mentioning sample points, the strategy is also important to understand. 

Fig-2.3 Data processing, Need to make a comparison of sites separately and in conjunction with the most suited environment for comparison and also in Spatio-temporal variations that are very important.  as rainfall is highly varied but temperature fluctuation is not in big range. 

A strong taxonomic recommendation is needed for comparison of species distribution, TAXonomic classification card may need to use for clarification. 

Reviewer 3 Report

Dear authors, the manuscript entitled "Spatial heterogeneity of root water conduction strategies of Zygophyllaceae plants in arid regions of China" addressed an important topic related to the resilience of the Loess plateau environment in China. The manuscript also shows the results of extensive fieldwork carried out by the research team. Unfortunately, despite the effort made by the authors, this manuscript requires extensive editing to improve the quality needed for the journal Biology. Therefore, I encourage the authors to improve this manuscript and resubmit it later.

Major Changes

- The introduction addressed different issues of high importance for the manuscript (e.g., drought effects, vessels anatomy); however, the order of ideas is too messy and does not help to identify the problem or the targeted region. Therefore, the authors should restructure the introduction by following a straightforward idea path procedure (e.g., the storytelling principle).

- Different sections within the introduction are more suitable for discussing the results (e.g., from lines 79 to 91).

- Two critical aspects of this manuscript are not present or identifiable in the text.: the research question and the aim of the study. 

- The "Materials and Methods" section has to be improved. For example, there is no information on the harvesting procedures, storage, length of samples, or sampling dates. Instead, the authors mention that "species with obvious taproots were selected in each sample" (Line 161). Can the authors extend the description of "obvious taproots" in an appendix with pictures or diagrams of the different plant types?

- The authors used correlation and probability values in different parts of the results. However, the Materials and Methods section only mentioned the statistical software used (i.e., SPSS). They did not provide information on the different statistical analyses they carried out or the accomplishment of assumptions. This omission makes it difficult to assess the statistics of the manuscript. 

-The authors skip any reference to the spatial distribution of the selected species. Table 1 should inform the reader about the chosen species in each site because it is key to understanding the previous results. Also, a dendrogram analysis may help the authors to improve the data analysis considering the measured anatomical variables and the species similarities among sites.

- There is a notorious wrong use of experimental design terminology. For example, the authors mention in line 160 the distribution of 11 samples from east to northwest, but those points are sampling sites, not samples. Consequently, there is an omission of how many samples per site per plant were collected. Therefore, I invite the authors to refine the following in the "Materials and Methods section":

-        Provide a more detailed description of geomorphology, pedology and geology, which, together with the climate and elevation, determine the main physiological characteristics that influence the water transport in plant roots for those 11 sites.

-        Describe how many plants per site were selected, as well as how many samples per plant were collected. In this way, the authors will guarantee the principle of replicability of scientific research.

-        The manuscript also analysis the hydraulic conductivity (from line 235 to line 238). The methodology is entirely theoretical, with no comparisons to experimental values obtained by other research on the same species. If the species lacks data, the authors should refer to the hydraulic conductivity of species in the area or biome to provide a better discussion.

The results section shows a clear description of the results. Nonetheless, the discussion is far behind the expected from such a study. The authors made broad assumptions about what can be inferred from their research. The most important part related to the variability and the impact of environmental drivers is not discussed at all in the manuscript.

Format Issues

- The reference numbering system used by the authors shows roman numbers; this should be changed.

- The first time a full species name is mentioned, it should include the authoring information of those species.

- The authors mixed several ways to show units in the form of kg m-1MPa-1s-1 (line 237) and μm/yr (e.g., line 368) or Age/yr (e.g., Figure 5). This shows the lack of consistency during the manuscript's formatting and analysis.

- Use of different fonts to immerse in the text (e.g., "precipitation" in line 153).

- Wrong formatting of equations (e.g., line 240, line 242, line 246).

- Table with non-descriptive captions (e.g., Table 1, Table 2, Table 3).

- Figures of poor quality (e.g., Figure 3, Figure 4).

Reviewer 4 Report

The manuscript has been thoroughly reviewed. This area of plant science is completely virgin and of global interest. The manuscript is well-written. The scientific approach is also satisfactory. All the sections are well presented. All the figures are also in good resolution, except Figure 3, which needs to be replaced with the better one. Minor english editing is also required to make the manuscript of more worth.

Round 2

Reviewer 1 Report

there are still a fare chances to improve its Language